# Role of β-Catenin Activation in the Tumor Immune Microenvironment and Immunotherapy of Hepatocellular Carcinoma

**DOI:** 10.3390/cancers15082311

**Published:** 2023-04-15

**Authors:** Masahiro Morita, Naoshi Nishida, Tomoko Aoki, Hirokazu Chishina, Masahiro Takita, Hiroshi Ida, Satoru Hagiwara, Yasunori Minami, Kazuomi Ueshima, Masatoshi Kudo

**Affiliations:** Department of Gastroenterology and Hepatology, Kindai University Faculty of Medicine, 377-2 Ohno-Higashi, Osaka-Sayama 589-8511, Japan

**Keywords:** hepatocellular carcinoma, β-catenin activation, immune microenvironment, immune checkpoint inhibitor, molecular target agent

## Abstract

**Simple Summary:**

The response rate to immune checkpoint inhibitor (ICI) monotherapy in hepatocellular carcinoma (HCC) is as low as 20%; therefore, it is important to identify the patient subgroups that respond to ICI. β-catenin activation is considered to be a key factor of ICI resistance. β-catenin activation is involved in immunologically cold tumor formation in the tumor microenvironment. Overall, controlling β-catenin activation may improve response rates of HCC tumors to ICI. β-catenin modulators and certain kinase inhibitors can suppress β-catenin activation; therefore, combining these drugs with ICIs is expected to further improve the therapeutic effect of ICIs in HCC.

**Abstract:**

Recently, the therapeutic combination of atezolizumab and bevacizumab was widely used to treat advanced hepatocellular carcinoma (HCC). According to recent clinical trials, immune checkpoint inhibitors (ICIs) and molecular target agents are expected to be key therapeutic strategies in the future. Nonetheless, the mechanisms underlying molecular immune responses and immune evasion remain unclear. The tumor immune microenvironment plays a vital role in HCC progression. The infiltration of CD8-positive cells into tumors and the expression of immune checkpoint molecules are key factors in this immune microenvironment. Specifically, Wnt/β catenin pathway activation causes “immune exclusion”, associated with poor infiltration of CD8-positive cells. Some clinical studies suggested an association between ICI resistance and β-catenin activation in HCC. Additionally, several subclassifications of the tumor immune microenvironment were proposed. The HCC immune microenvironment can be broadly divided into inflamed class and non-inflamed class, with several subclasses. β-catenin mutations are important factors in immune subclasses; this may be useful when considering therapeutic strategies as β-catenin activation may serve as a biomarker for ICI. Various types of β-catenin modulators were developed. Several kinases may also be involved in the β-catenin pathway. Therefore, combinations of β-catenin modulators, kinase inhibitors, and ICIs may exert synergistic effects.

## 1. Introduction

Hepatocellular carcinoma (HCC) is one of the most common malignancies and a leading cause of cancer-related mortality worldwide [1,2]. HCC exhibits a high diversity in carcinogens, pathological morphology, and genetic abnormalities. Additionally, treatment approaches for HCC differ according to variations in the number of tumors, tumor diameter, distribution, liver function, extrahepatic metastasis, and vascular invasion. However, it is estimated that approximately 50–60% of patients with HCC eventually receive systemic drug treatment [3,4]. In a recent study, nearly 80% of patients with HCC did not respond effectively to anti-PD-1 monotherapy, indicating that it is crucial to identify the subgroup that possess a good response to immune checkpoint inhibitors (ICIs) to effectively manage HCC [5,6,7,8]. According to a recent study, Wnt/β-catenin signaling may be a promising marker for predicting immune resistance to anti-PD-1 therapy in HCC [9,10,11,12]. Nonetheless, to understand the tumor responses and resistance to ICI, it is necessary to understand the characteristics of HCC from an immune microenvironment perspective.

## 2. Immune Exclusion Associated with Activation of the β-Catenin Pathway according to Basic Studies

It is clear that the effects of cancers on the immune microenvironment differ depending on the carcinogenic pathway [11,12,13,14,15,16,17]. The associations between different immunological microenvironments and carcinogenic pathways suggest that therapeutic responses to ICIs may also differ. Inappropriate activation of the Wnt/β-catenin pathway is believed to be involved in carcinogenesis. Specifically, there are multiple genetic abnormalities involved in the activation of the Wnt/β-catenin pathway; nonetheless, *CTNNB1* mutation is a typical driver mutation that is found in approximately 30% of HCC cases [18].

In 2015, Spranger et al. reported that Wnt/β-catenin pathway activation inhibits cytotoxic T cell infiltration in the immune microenvironment of malignant melanoma, resulting in resistance to immune checkpoint inhibitors [11]. Further investigation revealed that Batf3-dependent CD103-positive dendritic cell–derived chemokines are essential for the invasion of effector T cells into the tumor microenvironment. Wnt/β-catenin signaling was shown to inhibit this dendritic cell invasion into the tumor [19].

In 2019, Galarreta et al. reported the role of β-catenin pathway activation in HCC [12]. According to HCC mouse models, activation of β-catenin pathway was determined to inhibit dendritic cell recruitment in the tumor microenvironment, resulting in inhibition of T cell activation and reduction in CD8-positive T cell infiltration. Interestingly, CCL5-overexpressing β-catenin-activated mouse models were determined to possess restored immunosurveillance, delayed tumor progression, and an increased frequency in specific dendritic cells and CD8-positive cells. Responsiveness to ICI was also evaluated; antigen expression in tumor cells was determined to be important for the efficacy of PD-1 antibodies in conventional HCC mouse models. In a β-catenin-activated mouse model, the antitumorigenic effect of the PD-1 antibody was not observed regardless of the antigen expression status of the tumor cells. In other words, an HCC model with β-catenin activation exhibited poor immune cell infiltration; consequently, the antitumorigenic effect of the PD-1 antibody was difficult to acquire.

## 3. Relationship between β-Catenin Activation and ICI Effects in Human HCC

Basic research in mice revealed that immune exclusion is associated with β-catenin activation. Associations between β-catenin activation and the immune microenvironment were also reported in human HCC.

In 2020, we examined the association between oncogenic pathways in 154 resected human HCC specimens. We reported a relationship between PD-L1 expression in HCC and eight carcinogenic pathways (β-catenin, p53/cell cycle control, chromatin remodeling, epigenetic regulation, PI3K–Akt, oxidative and ER stress, DNA repair, and TERT promoter) based on immunohistochemical and genetic analysis [20].

Among the HCC variants, those with genetic mutations that lead to β-catenin activation had significantly lower CD8-positive cell infiltration and PD-L1 expression (*p* < 0.01 and *p* = 0.03, respectively). Therefore, it is postulated that the PD-1/PD-L1 antibody response is difficult to achieve.

There were also reports on the therapeutic efficacy of ICI in human HCC. Harding et al. reported response rates in 27 patients with HCC treated with ICIs (including anti-PD-1 antibody monotherapy, anti-PD-L1 antibody monotherapy, anti-CTLA antibody monotherapy, and ICI-ICI). Progressive disease (PD) was found across all 10 cases that possessed genetic mutations that activated the Wnt/β-catenin pathway [9]. On the other hand, in patients with HCC that did not possess genetic mutations that activated the Wnt/β-catenin pathway, one case of complete response (CR), two cases of partial response (PR), ten cases of stable disease (SD), and four cases of PD were observed. Furthermore, progression-free survival (PFS) was significantly shorter in the group with Wnt/β-catenin pathway–activating mutations than in the group without these mutations (2.0 vs. 7.4 months; *p* < 0.0001).

In 2021, we investigated the background factors and prognosis of 34 patients with unresectable HCC that were treated with anti-PD-1 antibody monotherapy by assessing pre-treatment tumor specimens [10]. Overall, β-catenin pathway activation was observed in 14 of 34 patients (41.2%). Background factors included (1) negative for β-catenin activation (β-catenin/GS staining negative), (2) PD-L1-combined positive score (CPS) ≥ 1, and (3) high infiltration of CD8-positive cells that significantly prolonged both PFS and overall survival (OS). CPS was defined as the number of PD-L1-positive cells (tumor cells, lymphocytes, and macrophages) divided by the total number of viable tumor cells × 100; CPS of PD-L1 (PD-L1-CPS) was calculated as previously reported [6]. The three factors established in this study were defined as positive prognostic factors, and stratified analysis was performed. Corresponding analysis was performed by dividing the patients into three groups: (1) without positive prognostic factors, (2) with 1 positive prognostic factor, and (3) with 2–3 positive prognostic factors. The prognosis of these patients was determined to significantly improve as the number of prognostic factors increased, according to PFS (*p* < 0.0001) and OS (OS *p* = 0.0048).

According to the above results, β-catenin activation is considered to be involved in ICI resistance in HCC. Furthermore, the effects of ICI are thought to be low in tumor immune microenvironments that possess poor CD8-positive cell infiltration and PD-L1 expression. Nonetheless, these studies evaluated only a small number of cases; therefore, further research with a larger sample size is required.

## 4. Relationship between β-Catenin Mutation and Immune-Related Gene Expression

In order to further investigate the role of β-catenin activation, we performed genetic analysis in groups with and without a β-catenin activating mutation (*CTNNB1* mutation). The corresponding expression statuses of the immune-related gene groups were then compared [10]. An ‘immunogram’ was used for comparison to visualize the expression of immune-related genes [21]. The immunogram consists of 10 axes; however, the CTNNB1 mutation group tended to possess low expression of immune-related genes, with a significantly lower expression in priming and activation, interferon (IFN)γ responses, inhibitory molecules, and inhibitory cells (Tregs). Additionally, IFNγ response–related genes were determined to be independent factors in multivariate analysis (*p* = 0.0490). Previous studies reported that IFNγ is an important cytokine in the immune response against tumors, and that PD-L1 expression is induced by IFNγ [22]. The following mechanism is considered to describe the IFNγ-dependent induction in PD-L1. Activated T cells in the tumor microenvironment release IFNγ and activate the JAK1/2-STAT1/3 pathway through IFNγ receptors (IFNγR1/2) on cancer cells. Then, STAT1/3 induces the expression of IFN regulatory factor 1 (IRF1); finally, IRF1 binds to the promoter region of PD-L1 to enhance PD-L1 expression [23].

## 5. Role of β-Catenin Mutation in the Classification of Immune Subclasses

### 5.1. Immune Subclass of HCC According to Pan-Cancer Analysis

In 2018, Thorsson et al. reported several immune subclasses across various cancers [24]. They performed an integrated analysis of 33 carcinomas and over 10,000 tumors using TCGA data; these tumors were then clustered according to their corresponding tumor immune microenvironments. Specifically, they were classified into six clusters. These clusters were characterized by differences in the macrophage or lymphocyte signatures, Th1:Th2 cell ratio, degree of intratumoral heterogeneity, aneuploidy, neoantigen load, cell proliferation, immunoregulatory genes, and prognosis. Cluster1 (wound healing) exhibited increased angiogenic gene expression, high proliferative ability, and a higher proportion of Th2 cells. Cluster2 (IFN-γ dominant) possessed a relatively high M1 macrophage polarization, high frequency of CD8-positive T cells, and relatively diverse T cell receptors. Cluster3 (inflammatory) exhibited a relatively high Th17 and Th1 gene expression. In contrast, somatic cell copy number changes and aneuploidy levels were lower in Cluster3 than those in other clusters. Cluster4 (lymphocyte depleted) was associated with a pronounced macrophage signature and suppression of Th1 cells. In addition, the rates of *CTNNB1*, epidermal growth factor receptor (*EGFR*), and *IDH1* mutations were high in Cluster4. Cluster5 (immunologically quiet) exhibited a particularly low lymphocytic infiltration and a relatively high percentage of M2 macrophages. Cluster6 (TGF-β dominant) was characterized by relatively high transforming growth factor-β (TGF-β) signature, with homogeneous and abundant Th1 and Th2 infiltration. Cluster3 (Inflammatory) and Cluster4 (Lymphocyte depleted) were the predominant clusters observed in HCC tumors. Additionally, specific driver mutations were reported to be correlated with leukocyte levels. Driver mutations that correlated with low leukocyte levels in all carcinomas were *CTNNB1*, *NRAS*, and *IDH1*; driver mutations that correlated with high leukocyte levels were *BRAF*, *TP53*, and *CASP8*.

### 5.2. HCC Immune Subclasses and β-Catenin Mutation

In 2017, Sia et al. described an “immune class” (herein referred to as the “inflamed class”) following gene expression analysis of 956 HCC cases. Approximately 25% of HCCs belong to the inflamed class. Specifically, the inflamed class is characterized by severe lymphocytic infiltration and elevated PD-1 and PD-L1 expression [25,26]. Approximately 60–80% of HCC tumors can be categorized as “non-inflamed class”; this class can then be further divided into two groups based on the corresponding mechanisms of immune escape. The two groups include the intermediate subclass and the excluded subclass, which is characterized by the *CTNNB1* mutation and possesses poor immune cell infiltration.

Kurebayashi et al. histologically classified the immune microenvironment using immunostaining of 919 regions in 158 HCC cases [27]. According to this analysis, the immune microenvironment can be classified as follows: (1) Immune-high: characterized by the co-infiltration of extrafollicular B cells and plasma cells, in addition to CD4, CD8-positive T cell infiltration. (2) Immune-mid: moderate T-cell infiltration but poor B cell and plasma cell infiltration. (3) Immune-low: poor immune cell infiltration. In this study, most HCCs with β-catenin activation were classified as the immune-low group. (immune-high N = 2; immune-mid N = 3; immune-low N = 30).

Fujita et al. evaluated immune-related genes in 234 HCCs cases and classified these HCC tumors based on local immune responses [28]. Overall, they were classified based on their immunosuppressive mechanism as follows: (1) CTNNB1 group (activation of Wnt/β-catenin signaling), (2) Treg group (regulatory T cell expression), (3) TAM group (tumor-associated macrophage M2 expression), and (4) high cytotoxicity group (high immune cytotoxicity but no immunosuppressive mechanisms). In this report, inflamed HCCs were subclassified into the high cytotoxicity and Treg groups, whereas non-inflamed HCCs were subclassified into the TAM and CTNNB1 groups. These immunosuppressive mechanisms are strongly associated with mutations in *CTNNB1* and *ARID2* in cancer cells.

### 5.3. New Inflamed Subclass: Relationship between Immune-Like Subclass and CTNNB1 Mutation

Montironi et al. analyzed 240 HCC cases and proposed an immune-like subclass within the inflamed class [29,30]. The immune-like subclass has immunological characteristics similar to those of other inflamed subclasses, such as high expression of immune checkpoint molecules and enhanced inflammation-related pathways. In contrast, the proportions corresponding to the Hoshida S2 and Chiang CTNNB1 classes were significantly higher in the immune-like subclass than in the other inflamed subclasses. In addition, the rate of β-catenin activation was significantly higher in the immune-like subclass than that in other inflamed subclasses (54% vs. 3%; *p* = 2.35 × 10^−6^). Regarding non-inflamed class, the excluded subclass had the lowest tumor immune cell infiltration and a higher incidence of *CTNNB1* mutations than the intermediate subclass (93% vs. 17%; *p* = 5.55 × 10^−12^).

### 5.4. Relationship between Nonalcoholic Steatohepatitis (NASH)/Nonalcoholic Fatty Liver Disease (NAFLD) HCC and β-Catenin Mutation

In 2021, Pfister et al. reported that ICIs had an inferior effect on NASH-HCC even when lymphocyte infiltration was observed. NASH-specific CD8-positive cells induce hepatocellular injury and fibrosis, which results in the promotion of hepatocarcinogenesis [31,32]. Additionally, Pinyol et al. reported the molecular characteristics of NASH-HCC. The corresponding analysis indicated that the frequency of *ACVR2A* mutations was higher in NASH-HCC than in non-NASH-HCC. However, with regard to the molecular class, NASH-HCC was significantly less likely to belong to the CTNNB1 subclass than non-NASH-HCC (16% vs. 31%; *p* = 0.02) [33,34]. In another report, *CTNNB1* mutation was determined to be a dominant event in the coding DNA sequence of NAFLD-HCC (33%). Immune exclusion correlated with *CTNNB1* mutations in patients with NAFLD-HCC Specifically, *CTNNB1* mutations lead to immune exclusion via the upregulation of TNFRSF19, which subsequently represses senescence-associated secretory phenotype (SASP)-like cytokines (including IL6 and CXCL8). Nonetheless, this phenomenon can be reversed by using the Wnt-modulator ICG001 [35].

## 6. Factors Affecting the Wnt/β-Catenin Pathway

Basic research revealed that the Wnt/β-catenin pathway is complex and involves various molecules [36,37,38,39,40,41,42,43,44,45,46,47,48] (Figure 1). Abnormalities in the components of the Wnt/β-catenin pathway can lead to the corresponding activation of this pathway. Several molecules were reported to indirectly contribute to β-catenin activation by affecting key components of the Wnt/β-catenin pathway. The specific molecules that act on key components of the Wnt/β-catenin pathway are detailed in Figure 1.

### 6.1. Molecules Acting on Wnt-Frizzled Complex and LPR5/6

Tumor-associated macrophages (TAMs) are major components of the tumor immune microenvironment and play an important role in the progression of HCC. Tumor cell–derived Wnt ligands stimulate M2-like polarization of TAMs through canonical Wnt/β-catenin signaling, resulting in tumor growth and immunosuppression. [49]. At present, 19 types of Wnt were identified in mammals. Wnts with strong transforming abilities, such as Wnt3a, activate the Wnt/β-catenin pathway, whereas Wnts with weak transforming ability, such as Wnt5a, are thought to activate non-canonical Wnt signaling. It was reported that tyrosine kinase–like orphan receptor 2 directly binds Wnt5a, resulting in the inhibition of Wnt/β-catenin signaling and activation of non-canonical Wnt signaling [50,51]. CDK/cyclin Y phosphorylates low density lipoprotein receptor-related protein 5/6 (LRP5/6) and activates mitotic Wnt/β-catenin signaling [52,53]. Src reduces LRP6 levels on the cell surface and interferes with LRP6 signalosome formation [54]. Furthermore, fibroblast growth factor 2 (FGF2) uses the ERK/MAP kinase pathway to phosphorylate LRP6. FGF2 activates WNT/β-catenin signaling via LRP6 phosphorylation [55]. Src phosphorylates frizzled, leading to Fyn recruitment and activation. Finally, β-catenin is released from the junctional complex by Fyn [56].

### 6.2. Molecules Acting on GSK3β

X protein is a hepatitis B virus (HBV)-encoded oncogenic protein that promotes HCC progression. HBV X protein (HBx) interacts with non-muscle myosin heavy chain IIA (MYH9) and induces its expression by modulating GSK3β/β-catenin/c-Jun signaling. Alternatively, silencing MYH9 blocks HBx-induced GSK3β ubiquitination, thereby activating the β-catenin destruction complex [57]. Elevated levels of polymeric immunoglobulin receptor (pIgR) were found in circulating extracellular vesicles (EVs) in patients with HCC. EVs enriched with pIgR consistently promote cancer stemness and cancerous phenotypes. EV-pIgR-induced cancer aggressiveness was observed to be reversed by Akt and β-catenin inhibitors, confirming that EV-pIgR activates PDK1/Akt/GSK3β/β-catenin signaling [58]. Lemur tyrosine kinase 2 (LMTK2) is a member of the lemur family of kinases that plays a key role in cell physiology and disease pathogenesis. Silencing LMTK2 suppress HCC progression through modulation of GSK-3β/Wnt/β-catenin signaling. Furthermore, LMTK2 is thought to be involved in GSK-3β phosphorylation and β-catenin protein expression [59].

### 6.3. Molecules Acting on Axin

GSK3β phosphorylates Axin1 to facilitate the ubiquitination and proteasomal degradation of β-catenin [60]. IFN induces expression of IFN-stimulated genes (ISGs). As an innate immune effector, ISG12a (also known as IFI27) promotes an innate immune response to viral infections. ISG12a promotes cancer immunity by suppressing the canonical Wnt/β-catenin signaling pathway. Specifically, ISG12a promotes β-catenin proteasomal degradation by inhibiting the degradation of ubiquitinated Axin, thereby suppressing the canonical Wnt/β-catenin signaling pathway [61].

### 6.4. Molecules Acting on Dvl

Casein kinase 1 (CK1) phosphorylates disheveled (Dvl) and promotes Dvl signaling [62]. It was also reported that the phosphorylation of Dvl by CK1 triggers a negative feedback loop that inhibits Wnt/β-catenin signaling [63]. Dvl is involved in centrosome cycle regulation; specifically, Dvl accumulates during the cell cycle and is associated with NIMA-related kinase 2 (NEK2). Centrosome-accumulated Dvl is phosphorylated by NEK2. This phosphorylation releases Dvl from the centrosome and increases activation of Wnt/β-catenin signaling [64,65]. After Wnt3a stimulation, Src binds and activates Dvl, phosphorylating Dvl and β-catenin and resulting in nuclear accumulation of β-catenin [66].

### 6.5. Molecules Acting on Adenomatous Polyposis coli (APC)

GSK3β phosphorylates APC to facilitate the ubiquitination and proteasomal degradation of β-catenin [67,68,69]. CK1 also phosphorylates APC to further facilitate this ubiquitination and proteasomal degradation of β-catenin [70,71,72,73].

### 6.6. Molecules Acting on βcatenin and T-Cell Factor (TCF)/Lymphoid Enhancer Factor (LEF)

GSK3β phosphorylates β-catenin, promoting its ubiquitination and proteasomal degradation [67,68,69]. CK1 also phosphorylates β-catenin to promotes its ubiquitination and proteasomal degradation. Additionally, CK1 phosphorylates Transcription factor 3 (TCF3), which promotes its interaction with β-catenin [62,63,74]. The phosphorylation of LEF-1 by CK2 significantly enhances its affinity for β-catenin and stimulates the transactivation of the β-catenin/LEF-1 complexes [75,76].

Mesenchymal epithelial transition factor (MET) phosphorylates β-catenin on tyrosine residues, leading to the dissociation of β-catenin from MET and corresponding nuclear translocation of β-catenin [77]. Fibroblast growth factor receptor 2 (FGFR2), FGFR3, and EGFR were reported to directly phosphorylate β-catenin at tyrosine residue 142. This phosphorylation results in β-catenin release from membrane junctions and an increase in cytosolic β-catenin [55]. p21-activated kinase 4 (PAK4) interacts with and phosphorylates β-catenin on Ser675, thereby promoting TCF/LEF transcriptional activity and stabilizing β-catenin by inhibiting its degradation [78]. Src docks to the C-terminus of Dvl2 and enhances Wnt3a-stimulated TCF/LEF-dependent transcription [66]. Further, TAK1 activation promotes NLK activity. NLK then phosphorylates TCF and interferes with the binding of β-catenin to TCF target sites [79]. Salt-inducible kinase 1 (SIK1) phosphorylates the silencing mediators retinoic acid and thyroid hormone receptor (SMRT) at threonine residue 1391, thereby promoting the translocation of SMRT into the nucleus. Then, phosphorylated SMRT recruits the nuclear receptor core-pressor/histone deacetylase 3 corepressor complex to β-catenin/TCF. This results in the inhibition of Wnt target gene transcription. Therefore, loss of SIK1 leads to activation of the Wnt/β-catenin signaling [80].

Sia et al. reported that the CTNNB1 class of HCC tumors exhibits overexpression of protein tyrosine kinase 2 (PTK2) signaling [25]. PTK2, also known as focal adhesion kinase (FAK), activates Wnt/β-catenin signaling and promotes cancer stem cell characteristics. PTK2 reduces β-catenin degradation and increases the nuclear accumulation of β-catenin [81]. Alternatively, FAT10 was implicated in cell growth and survival. FAT10 and homeobox B9 (HOXB9) each promote HCC metastatic progression. Consequently, silencing of FAT10 decreased HOXB9 expression and inhibited HCC invasion and metastasis. Specifically, FAT10 regulates the β-catenin/TCF4 pathway by directly binding to β-catenin and preventing its ubiquitination and degradation, thereby regulating HOXB9 expression [82].

## 7. Development of Therapeutic Agents Targeting the β-Catenin Pathway

### 7.1. Small Molecule Wnt Pathway Inhibitor

The broad involvement of Wnt signaling in cancer progression drove extensive research efforts to target the Wnt pathway using small molecules (Table 1).

IWP is a specific small-molecule inhibitor of the Wnt pathway that blocks Porcupine, an enzyme that promotes the acylation of Wnt proteins. The IWP2 compound can inactivate Porcupine with a high degree of selectivity [36,83,84]. Alternatively, XAV939 stimulates β-catenin degradation by stabilizing Axin, the concentration-limiting component of the destruction complex. Axin stability is regulated, in part, by ADP-ribosylation catalyzed by tankyrase. Specifically, XAV939 stabilizes Axin by inhibiting the poly ADP-ribosylating enzymes tankyrase 1 and tankyrase 2 [85]. IWR (IWR-1/2) are small molecules that antagonize Wnt signaling by stabilizing the Axin destruction complex [36,84]. β-catenin stability is also controlled by phosphorylation, which is, in part, regulated by CK1α. CK1α appears to be activated by pyrvinium, resulting in the inhibition of the Wnt signaling pathway [86]. The formation of a complex between β-catenin and TCF/LEF is a crucial step in Wnt signaling. To generate a transcriptionally active complex, β-catenin recruits the transcriptional coactivators cyclic AMP response element-binding protein (CBP) or its closely related homolog, p300, and other components of the basal transcription machinery. ICG-001 is a selective, low-molecular-weight inhibitor that antagonizes β-catenin/TCF-mediated transcription. ICG-001 binds specifically to CBP and disrupts the interaction of CBP with β-catenin [87]. To inhibit Wnt signaling, an effective target would be the complex between TCF and β-catenin. Consequently, small-molecule antagonists of the oncogenic TCF/β-catenin protein complex, such as PKF115-584, were suggested [88]. E7386 is a selective inhibitor of the interaction between β-catenin and CBP. E7386 induced the infiltration of CD8-positive cells into tumor tissue. Furthermore, in combination with anti-PD-1 antibody, it demonstrated synergistic anti-tumor activity against mouse mammary tumors developed in mouse mammary tumor virus-Wnt1 transgenic mice [89].

### 7.2. The Potential of Kinase Inhibitors to Improve the Tumor Immune Microenvironment from Immunologically Cold to Hot

As previously discussed, β-catenin activation participates in the formation of an immunologically cold tumor with poor immune cell infiltration. Some kinase inhibitors may modulate β-catenin activation and have the potential to alter the tumor immune microenvironment from cold to hot (Table 2).

MET inhibitors may theoretically be effective in regulating the β-catenin pathway. MET inhibitors include cabozantinib, tepotinib, golvatinib, and capmatinib. Furthermore, cabozantinib is used for advanced hepatocellular carcinoma based on corresponding clinical trial results [90].

EGFR, FAK, and Src inhibitors may theoretically affect the β-catenin pathway. EGFR inhibitors include erlotinib, gefitinib, lapatinib, cetuximab. Additionally, defactinib inhibits FAK and saracatinib inhibits Src.

To date, vascular endothelial growth factor (VEGF) was not shown to be involved in β-catenin activation. Nonetheless, VEGF contributes to tumor angiogenesis and effects the tumor microenvironment by inducing the infiltration of various cells, including Tregs, TAMs, and myeloid-derived suppressor cells (MDSCs), and inducing the release of immunosuppressive cytokines. Therefore, VEGF inhibition may improve the tumor microenvironment. It was reported that VEGF inhibition reduces TAMs and Tregs in the tumor microenvironment, decreases TGF-β and IL-10 expression, decreases the expression of T-cell exhaustion markers, such as PD-1 and TIM-3, and increases the release of immunostimulatory cytokines [91].

β-catenin activation and FGFR4 expression were also reported to be positive correlation in hepatocellular carcinoma [92]. Lenvatinib, a molecular target agent, was reported to be highly selective for kinases that are important for angiogenesis, such as vascular endothelial growth factor receptor (VEGFR) and FGFR; therefore, lenvatinib may be a promising therapeutic candidate [93].

## 8. Potential Synergistic Effects of Combining Kinase Inhibitors and Wnt/β-Catenin Inhibitors with ICIs

According to prior basic research, the following mechanism was suggested for the β-catenin-dependent regulation of the tumor immune microenvironment. β-catenin activation reduces the production of chemokines such as CCL5, which inhibits the recruitment of CD103+ dendritic cells and prevents antigen presentation by dendritic cells. As a result, T-cell activation is inhibited and tumor infiltration of CD8-positive cells is prevented [11,12,19].

Considering the corresponding results in human HCC, one possible hypothesis regarding the relationships between β-catenin pathway activation, CD8-positive cells infiltration, PD-L1 expression, IFNγ, and ICI efficacy are as follows. In HCCs with β-catenin activation, dendritic cells cannot be recruited to the tumor, which ultimately prevents antigen presentation and T cell activation. Consequently, immune exclusion occurs, and IFNγ is not released because CD8-positive cells do not target the tumor without detecting antigen presentation by the dendritic cells [9,10,20]. Furthermore, PD-L1 expression is not induced because of the corresponding reduced levels of IFNγ; therefore, the effectiveness of ICIs reduces [22,23]. As a result, activation of the Wnt/β-catenin pathway causes immune exclusion. This is a major factor that contributes to the development of primary resistance to ICIs (Figure 2). Nonetheless, not all cases of β-catenin-activated HCC possess low CD8-positive cells infiltration or PD-L1 expression; therefore, not all ICI resistances can be explained by a single mechanism. Nevertheless, β-catenin activation may be an important factor in shaping the tumor immune microenvironment.

Therefore, β-catenin activation may serve as a biomarker to predict the efficacy of ICI treatment. According to Montironi et al.’s classification, *CTNNB1* mutations are found not only in immune-excluded subclasses but also in the inflamed immune-like subclass [29]. Alternatively, in our previous report, we established an ICI-resistant subgroup with a combination of low PD-L1 expression, low CD8-positive cells, and β-catenin activation, which appeared to confer immune exclusion [10]. Furthermore, Montironi et al. suggested that 8q24.3 amplification and PTK2 hypomethylation are also factors in immune exclusion, and β-catenin activation may be a marker for immune exclusion that could be assessed in combination with these genetic changes.

Regulating the Wnt/β-catenin pathway may alter the tumor microenvironment from immunologically cold to hot. It may be possible to strongly inhibit the Wnt/β-catenin pathway by combining kinase inhibitors and Wnt/β-catenin inhibitors. If this β-catenin pathway can be controlled, there is a possibility that the efficacy of ICIs improved; therefore, combining the use of these kinase and Wnt/β-catenin inhibitors with ICIs may result in a synergistic effect. In recent years, clinical trials also began in combination with Wnt/β-catenin inhibitors, kinase inhibitors, and ICIs. A phase 2 study is ongoing to assess the objective response rate of E7386 in combination with pembrolizumab or of E7386 in combination with pembrolizumab plus lenvatinib in HCC [94].

## 9. Conclusions

According to prior studies, the tumor immune microenvironment can become “immunologically cold” following the activation of β-catenin; this immunologically cold tumor environment is important for the corresponding reduction in ICI response rate. The β-catenin pathway also plays an important role in immune subclassification. Specifically, it was reported that genetic mutations and various other factors, such as kinase activity, are involved in β-catenin activation. Regulating β-catenin pathway activation may improve the immune microenvironment; therefore, various β-catenin modulators are under development. Additionally, certain protein kinases are understood to indirectly act on the β-catenin pathway; these kinases are also expected to be utilized as novel targets for improving the immune microenvironment. The combined use of kinase inhibitors, ICIs, and β-catenin modulators may be an effective strategy for the treatment of HCC subtypes with β-catenin activation. Therefore, future reports regarding the development of these therapeutic strategies are expected.

## Figures and Tables

**Figure 1 cancers-15-02311-f001:**
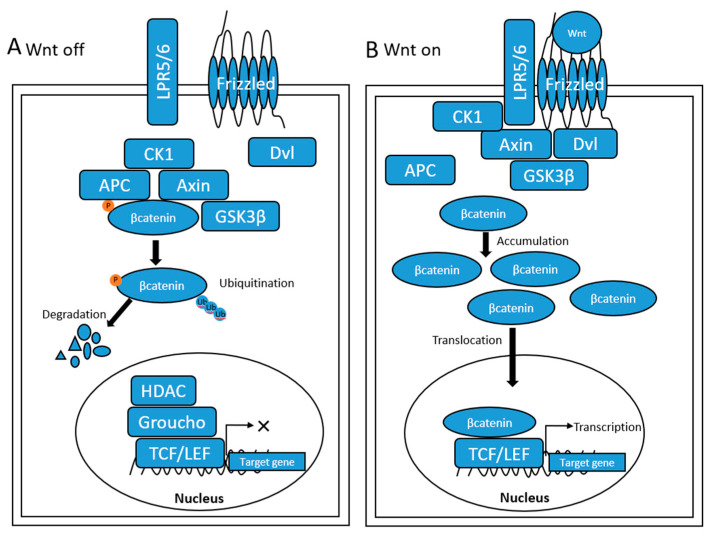
Canonical Wnt signaling pathway. (**A**) Wnt off. When Wnt is not bound to the frizzled/lipoprotein receptor-related protein (LRP) coreceptor complex, Wnt/β-catenin signaling is inactive. The destruction complex can then phosphorylate β-catenin, which leads to β-catenin ubiquitination and corresponding proteasomal degradation. In the nucleus, the binding of Groucho to T-cell factor (TCF) inhibits the transcription of Wnt target genes. (**B**) Wnt on. Binding of the WNT ligand to the frizzled-LRP5/6 coreceptor recruits disheveled (Dvl) and Axin to disrupt the destruction complex. This then causes the accumulation of β-catenin, which translocates into the nucleus. β-catenin then binds to TCF/lymphoid enhancer factor (LEF) in the nucleus and activates the transcription of Wnt target genes.

**Figure 2 cancers-15-02311-f002:**
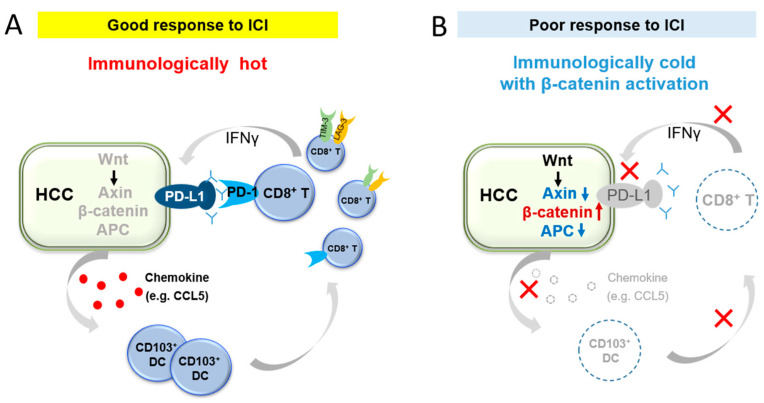
Schematic representing a hypothesis for β-catenin activation and immune response in the tumor microenvironment. (**A**). Hepatocellular carcinoma (HCC) cells produce chemokines (e.g., CCL5), which results in CD103+ dendritic cell recruitment and antigen presentation. As a result, T cell priming and activation occurs; then, CD8+ cells migrate to tumor cells. CD8+ cells target tumor cells and release IFNγ. IFNγ induces PD-L1 expression. Therefore, ICIs targeting the PD-1/PD-L1 axis function well in this tumor microenvironment. (**B**). HCC with β-catenin activation causes a decrease in certain cytokines (e.g., CCL5), which results in the inhibition of CD103+ dendritic cell recruitment. This inhibits T cell activation, forming a tumor microenvironment with poor CD8+ cell infiltration. Therefore, induction in PD-L1 expression is reduced due to reduced IFNγsecretion. Therefore, this tumor microenvironment is expected to be poorly responsive to ICIs that target the PD-1/PD-L1 axis.

**Table 1 cancers-15-02311-t001:** Small molecule Wnt pathway inhibitors.

Small Molecule	Molecular Target	Mechanism
IWP	Porcupine	IWP blocks Porcupine, the enzyme promoting acylation of Wnt proteins.
XAV939	Tankyrase/Axin	XAV939 stimulates β-catenin degradation by stabilizing Axin.
IWR	Axin	IWR (IWR-1/2) antagonize Wnt signaling by stabilizing the Axin destruction complex.
Pyrvinium	CK1	Pyrvinium appears to activate CK1, resulting in inhibition of the Wnt pathway.
ICG-001	CBP	ICG-001 binds specifically to CBP and antagonizes β-catenin/TCF-mediated transcription.
E7386	CBP/β-catenin	E7386 blocks the interaction between the exogenous N-terminal region of CBP and endogenous β-catenin
PKF115-584	TCF/β-catenin	Antagonist of the TCF/β-catenin protein complex.

CK1, casein kinase 1; CBP, cyclic AMP response element-binding protein.

**Table 2 cancers-15-02311-t002:** Therapeutic targets for improving the tumor immune microenvironment.

Target	Rationale	Inhibitors
MET	MET phosphorylates tyrosine residues of β-catenin, causing dissociation of β-catenin from MET and nuclear translocation of β-catenin.	Cabozantinib Tepotinib Golvatinib Capmatinib
EGFR	EGFR phosphorylates β-catenin at tyrosine residue 142, resulting in β-catenin release from membrane junctions and an increase in cytosolic β-catenin.	Erlotinib Gefitinib Lapatinib Cetuximab
FAK (PTK2)	FAK, also known as PTK2, reduces β-catenin degradation and increases the nuclear accumulation of β-catenin.	Defactinib
Src	Src phosphorylates β-catenin, resulting in the accumulation of β-catenin in the nucleus. This promotes TCF/LEF transcription.	Saracatinib Dasatinib
VEGF	VEGF was not proven to be involved in β-catenin activation. Nonetheless, VEGF is involved in tumor angiogenesis and affects the tumor microenvironment by inducing Tregs, TAMs, and MDSCs and inducing immunosuppressive cytokine release. VEGF inhibitors may improve the tumor microenvironment.	Bevacizumab Ramucirumab Aflibercept Beta Lenvatinib Sorafenib Regorafenib
FGFR	FGFR2 and FGFR3 phosphorylate β-catenin at tyrosine residue 142, which leads to the release of β-catenin from membrane junctions and an increase in cytoplasmic β-catenin.	Pemigatinib Futibatinib Infigratinib Lenvatinib

MET, mesenchymal epithelial transition factor; EGFR, epidermal growth factor receptor; FAK, focal adhesion kinase; PTK2, protein tyrosine kinase 2; VEGF, vascular endothelial growth factor; TAMs, tumor-associated macrophages; MDSCs, myeloid-derived suppressor cells; FGFR, fibroblast growth factor receptor.

## Data Availability

Not applicable.

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
