# Peer review of "Role of β-Catenin Activation in the Tumor Immune Microenvironment and Immunotherapy of Hepatocellular Carcinoma"

_cancers, 2023, doi:10.3390/cancers15082311_

Round 1
Reviewer 1 Report
In present study, Morita et al reviewed recent progress of b-catenin activation and response of immune therapy in human HCC. The author summarized new discoveries related to b-catenin activation and microenvironment regulation. Meanwhile, the author reviewed effects of modulating b-catenin activation and response in immunotherapy which is important for the field. However, authors failed to offer an in-depth view of the achievement in the past and offer some insights on the future directions. More importantly, the current submitted manuscript need logically reorganized and it should not just only enumerate research findings nor reads like many paper pieced into this review.
At last, language of the current need intensively editing and it is really hard to understand. Here are some examples:
eg. line 23-25,Among them, basic studies have shown that Wnt/β catenin pathway activation causes "immune exclusion" that is
poor in CD8 positive cells infiltration.
line 31-32, It is also possible that several kinases are involved in the β-catenin pathway.
line 49-50,To understand response and resistance to ICI, it is necessary to understand the characteristics of HCC from the perspective of the immune microenvironment.
This reviewer would highly suggest the author reorganize their manuscript and improve the language for reconsideration.
Author Response
Thank you for your detailed feedback.
We have revised the text to better convey our intentions.
In addition, English proofreading was also carried out.
eg.
line 23-25 Wnt/β catenin pathway activation causes "immune exclusion", associated with poor infiltration of CD8-positive cells.
line 31-32,  Several kinases may also be involved in the β-catenin pathway.
line 49-50, Nonetheless, to understand the tumor responses and resistance to ICI, it is necessary to understand the characteristics of HCC from an immune microenvironment perspective.
Reviewer 2 Report
The review entitled “Implication of β-catenin activation in tumor immune microenvironment and immunotherapy for hepatocellular carcinoma” by Morita et al. comprehensively summarizes potential HCC therapeutic effects by modulating the activation of the β-catenin pathway from a molecular biological and immunological perspective.
On a minor note, isn't â‘¢ on page 4 line 175, (3)?
Also, in Figure 1, A:Wnt absent and B:Wnt presence would be better to be A:Wnt off and B:Wnt on, as the signaling is on or off dependent on whether the Wnt ligand binds to the Frizzeled/LRP receptor or not? 
Author Response
Thank you for your detailed feedback.
We have corrected the part you pointed out.
In addition, English proofreading was also carried out.
P4L182  As you pointed out, it is (3), not ③.
Figure 1  As you pointed out, the description has been changed.
Round 2
Reviewer 1 Report
The author have addressed this reviewer's concerns regarding to the language and style, the current version is now acceptable.